# The Role of α-Synuclein Oligomers in Parkinson’s Disease

**DOI:** 10.3390/ijms21228645

**Published:** 2020-11-17

**Authors:** Xiao-yu Du, Xi-xiu Xie, Rui-tian Liu

**Affiliations:** 1National Key Laboratory of Biochemical Engineering, Institute of Process Engineering, Chinese Academy of Sciences, Haidian District, Beijing 100190, China; xydu19@ipe.ac.cn (X.-y.D.); xxxie@ipe.ac.cn (X.-x.X.); 2School of Chemical Engineering, University of Chinese Academy of Sciences, Beijing 100049, China

**Keywords:** α-synuclein, Parkinson’s disease, oligomers, structure, toxicity

## Abstract

α-synuclein (α-syn) is a protein associated with the pathogenesis of Parkinson’s disease (PD), the second most common neurodegeneration disease with no effective treatment. However, how α-syn drives the pathology of PD remains elusive. Recent studies suggest that α-syn oligomers are the primary cause of neurotoxicity and play a critical role in PD. In this review, we discuss the process of α-syn oligomers formation and the current understanding of the structures of oligomers. We also describe seed and propagation effects of oligomeric forms of α-syn. Then, we summarize the mechanism by which α-syn oligomers exert neurotoxicity and promote neurodegeneration, including mitochondrial dysfunction, endoplasmic reticulum stress, proteostasis dysregulation, synaptic impairment, cell apoptosis and neuroinflammation. Finally, we investigate treatment regimens targeting α-syn oligomers at present. Further research is needed to understand the structure and toxicity mechanism of different types of oligomers, so as to provide theoretical basis for the treatment of PD.

## 1. Introduction

α-syn is a 140-amino-acid protein mainly located at presynaptic terminals and is abundant in the brain [1]. The physiological role of α-syn remains unclear. Some evidence suggests that physiological concentration of α-syn can regulate the amount of SNARE complex, which regulates the release of neurotransmitters [2]. However, excessive accumulation of α-syn has been linked with Parkinson’s disease (PD), the second most common neurodegenerative disease. Triplication of α-syn gene locus, which increases the level of α-syn, is associated with increased risk of PD [3]. Some mutations of α-syn, such as A53T and A30P, can also lead to PD [4,5]. These mutations inhibit the degradation of α-syn, resulting in α-syn accumulation [6].

When the balance between the production and clearance of α-syn is disturbed, the soluble monomeric α-syn aggregates and misfolds into oligomers, then amyloid fibrils and finally Lewy body [7]. Lewy body is one of the pathological hallmarks of PD, and the relevance between Lewy body and PD was revealed more than 30 years ago [8]. However, the exact pathogenesis of PD remains elusive. In the last decade, more and more evidence suggests that the oligomeric form of α-syn plays a vital part in the pathogenesis of PD [9,10,11,12]. In this review, we describe the current understanding of the structure, neurotoxic mechanisms, seed effects and propagation of α-syn oligomers, as well as therapies targeting oligomeric α-syn.

## 2. Structure of α-syn Oligomers

α-syn can exist in various forms, from unfolded monomers to fibrils (Figure 1). Oligomers are macromolecular complexes formed by non-covalent binding of proteins. For α-syn, oligomers usually consist of several to dozens of monomers. That means α-syn oligomers are highly heterogeneous, from molecular weight to structure. Unlike fibrils, oligomers are soluble and usually unstable because they can change their conformations rapidly. While the structure of α-syn fibrils has been revealed at the atomic level by cryo-electron microscopy, we still lack a clear understanding of the structure of α-syn oligomers [13].

Under normal conditions, α-syn can bind to lipid membrane to perform physiological functions [2] or form a tetramer with α-helical structure that can resist abnormal aggregation [14]. It is believed that, in the abnormal aggregation process, some monomers form oligomers that can eventually become fibrils. These kinds of oligomers are also known as “on-pathway” species. Some oligomers cannot form fibrils, which are called “off-pathway” species [15]. Some kinds of oligomers have been observed during the aggregation process. A meta-stable α-syn oligomer species, which is a mostly disordered sphere with approximate 100-Å diameter and is made up of 11 monomers, can convert into either on-pathway or off-pathway species [16]. Before the β-sheet rich fibril is formed, there is a helix-rich intermediate, which is cytotoxic and can promote aggregation of α-syn [17]. When helix-rich oligomers are transformed into fibrils, the α-helical secondary structures decrease and, in contrast, the β-sheet structures increase [18]. However, not all oligomers can elongate fibrils. Two oligomers have been reported to be this type. Their amount of β-sheet is between the disordered monomers and the β-sheet rich fibrils, and they cannot form fibrils. Instead, they prevent the formation of amyloid fibrils [19].

Studies have shown that certain secondary structures are the structural basis for α-syn oligomers to exert cellular toxicity. Amphipathic α-helices in N-terminal regions help oligomers to bind the surfaces of the lipid bilayers, leading to cytotoxicity [20]. Therefore, antibody targeting N-terminal of α-syn greatly reduces the toxicity of oligomers [21]. Meanwhile, a structure containing β-sheet is also necessary to generate toxicity. The core of oligomers which is rich in β-sheet structure can insert lipid bilayer and destroy membrane integrity [20]. Soluble α-syn oligomers dissociated from amyloid fibrils can form pores in lipid bilayers and their β-sheet structure is different from that of amyloid fibrils [22]. It is reported that α-syn oligomers exhibit an antiparallel β-sheet structure, while fibrils have parallel structure [23].

Although the exact structure of α-syn oligomers remains unknown, a low-resolution structure of α-syn oligomers that exist in the process of fibrillation has been solved (Figure 2A). The oligomer is a slightly elongated circular shape with a cavity in the middle, whose morphology was similar to that previously observed by electron microscopy [24]. Using cryo-electron microscopy, the structure of two subgroups of these α-syn oligomers that are kinetically trapped has been revealed. Despite the limited resolution and the heterogeneity of oligomers, these two subgroups of oligomers both have hollow cylindrical structures similar to some amyloid protofilaments (Figure 2B,C) [25]. To clearly understand the pathogenesis of PD, it is important to figure out which species of α-syn oligomers plays a critical role in the process of disease and parse the exact structure of the certain kind of oligomer. Further research is needed to solve this challenging problem.

## 3. Mechanism of α-syn Oligomer Toxicity

α-syn oligomers can produce cytotoxicity in a variety of ways, including mitochondrial dysfunction, endoplasmic reticulum (ER) stress, loss of proteostasis, synaptic impairment, cell apoptosis, and neuroinflammation (Figure 3).

### 3.1. Mitochondrial Dysfunction

Multiple studies have shown that α-syn oligomers can impair mitochondrial complex I. α-syn oligomers damage the respiration depending on complex I, inducing the selective oxidation of ATP synthase and mitochondrial lipid peroxidation, thereby promoting the opening of osmotic transition pore (PTP), leading to mitochondrial swelling and eventually cell death [26]. The impairment of mitochondrial complex I function caused by α-syn oligomers can also change membrane potential, destroy calcium homeostasis and promote the release of cytochrome C [27]. In addition, in an A53T mouse model, the α-syn oligomers were also shown to be associated with mitochondrial complex I damage [28].

In addition to damaging mitochondrial complex I, α-syn oligomers could induce mitochondrial dysfunction through other mechanisms. The high-affinity binding of α-syn oligomers and TOM20 peptide receptors prevent TOM20 from binding to its co-receptor TOM22, resulting in mitochondrial insufficiency of respiration and increased reactive oxygen species [29]. In cultured SH-SY5Y cells, α-syn oligomers destroy mitochondrial morphology, leading to organelle fragmentation [30]. It has proposed that α-syn oligomers, after being transported to dopaminergic (DA) neurons, damage mitochondria by activating mitochondrially encoded cytochrome c oxidase 2 [31]. A53T α-syn oligomers inhibit mitochondrial transport, which can be reversed by NAP (davunetide), indicating that microtubule damage is involved in PD pathology [32]. Besides, α-syn oligomers can disrupt the anterograde axonal transport of mitochondria by causing subcellular changes of the transport regulatory proteins and energy loss [33].

In addition to neurons, α-syn oligomers have also been observed to affect the mitochondrial function of astrocytes. The astrocytes can uptake α-syn oligomers and play a neuroprotective role. However, the long-term storage of α-syn oligomers in astrocytes can affect their mitochondrial integrity and make for neurotoxicity [34]. It has been shown that antibodies targeting α-syn oligomers can prevent α-syn accumulation and mitochondrial damage in cultured astrocytes [35]. Treatment of mouse astrocytes with different α-syn (monomer, oligomer and fiber) can activate astrocytes and increase the expression levels of oxidant and cytokines. Unlike α-syn monomers and fibers, only oligomers can induce mitochondrial dysfunction in astrocytes and significantly increase extracellular hydrogen peroxide production [36]. 

### 3.2. ER Stress

The ER is involved in protein synthesis, folding, modification and transport. When the ER’s ability to fold proteins reaches saturation, the stress occurs. α-syn accumulates in mitochondria before the onset of PD, forming toxic oligomers in vivo and causing ER stress [37]. This process continues with the progress of PD pathology, indicating that the ER stress triggered by oligomeric α-syn is involved in PD progression. Treatment with the ER stress inhibitor salubrinal could significantly reduce the disease manifestations in PD transgenic mouse model [37,38]. The ER stress response factor XBP1 (X-box binding protein 1), which has a protective effect, could be activated by α-syn oligomers, but not monomers or fibers, suggesting unique ability of oligomers to perturb cellular processes, including ER function [39]. Signal transduction between the ER and mitochondria via VAPB-PTPIP51 pathway regulates many neurological functions. α-syn oligomers can bind to VAPB, disrupt the VAPB-PTPIP51 tethers to weaken ER mitochondria associations, eventually cause ER stress [40]. Moreover, ER stress elicited by overexpression of α-syn can in turn increase the level of α-syn oligomers [41]. These studies indicate ER stress induced by α-syn oligomers is involved in PD pathology.

### 3.3. Loss of Proteostasis

The ubiquitin–proteasome system (UPS) and the autophagy–lysosomal pathway (ALP) are main pathways to clear the overexpressed or misfolded proteins in cells, which help maintain protein homeostasis. It has been reported that UPS is the main way to degrade α-syn. However, when UPS is heavily burdened, ALP also participates in the degradation process, which means that the two pathways both act in α-syn clearance [42].

α-syn oligomers can inhibit the function of proteasome, then form a vicious circle, which further intensifies the accumulation of misfolded proteins. Soluble α-syn oligomers are proved to impede the 20S and 26S proteasome activity, by blocking the entry of other proteasome substrates [43,44]. In an A53T mouse model, overexpressed α-syn led to 26S proteasome damage, followed by UPS dysfunction, which may promote neurodegeneration in vivo [45]. Parkin is an E3 ubiquitin ligase that plays a crucial part in ubiquitination. Exogenous α-syn oligomers can induce oxidation/nitrification stress, leading to parkin nitrosation, and parkin level reduction results in defective protein accumulation and cell death [46].

Lysosomal degradation pathway is also able to clear α-syn oligomers, and blocking the pathway causes the accumulation of α-syn oligomers and eventually cell death [47]. Overexpression of transcription factor EB (TFEB), an important regulator of the ALP, can rescue the dysfunction of lysosome. This protective effect is mediated by the elimination of α-syn oligomers [48]. These results show the interaction between α-syn oligomers and degradation pathways: the downregulation of UPS or ALP causes accumulation of α-syn oligomers, which in turn inhibits the clearance process. Promoting the removal of α-syn oligomers may be an effective way to restore the proteostasis and a potential therapeutic target of PD.

### 3.4. Synaptic Impairment

α-syn is abundant in synapse with an unclear physiological function. Under normal conditions, α-syn maintain the normal physiological function of synapses by binding to the SNARE-protein synaptobrevin-2/vesicle-associated membrane protein 2 (VAMP2), facilitating SNARE-complex assembly [49]. Other studies suggest that α-syn tends to form multimers with α-helical structure [14] and has the function to limit the movement of vesicles [50]. These suggest that α-syn can play different roles in maintaining synaptic homeostasis. However, α-syn oligomers with certain conformations can also impair synaptic function. Large α-syn oligomers were preferred in combination with VAMP2 to interfere the formation of SNARE complex, thus inhibiting the release of dopamine [51]. In an E57K mutant mouse model which is inclined to form oligomeric α-syn, loss of synapses and dendrites as well as decreased levels of synapsin 1 and synaptic vesicles were observed, and these led to further behavioral defects [52]. However, these changes were less obvious in wild-type α-syn mice. Another study using E46K and E57K mutant human iPSC-derived neurons showed a mechanism by which α-syn oligomers cause axonal dysfunction. Increased α-syn oligomers conduce to decreased axon density and synaptic degeneration through axon transport interruptions and energy defects, ultimately resulting in synapse loss [33]. This is consistent with a previous study that E57K variant α-syn oligomers dramatically impair axon transport process [12]. 

Several studies indicate that extracellular α-syn oligomers could do harm to synaptic transmission through targeting NMDA receptor. Rat hippocampal cells exposed to α-syn oligomers exhibit increased basal synaptic transmission through activating NMDA receptor and impaired long-term potentiation (LTP), which is the neurophysiological basis of learning and memory [53]. NMDA receptor could also be activated by the complex of α-syn and cellular prion protein (PrP^C^). This process triggers synaptic damage by disrupting calcium homeostasis and membrane integrity [54]. Besides, α-syn oligomers rather than fibrils can induce synaptic dysfunction and visual spatial memory impairment in the striatum by targeting GluN2A-NMDA receptors [55]. These studies suggest that α-syn oligomers promote the onset of PD by impairing synapses.

### 3.5. Cell Apoptosis

In addition to the above mechanisms of toxicity, α-syn oligomers can cause cell apoptosis. α-syn oligomers can induce the production of reactive oxygen species (ROS) dependent on free metal ions, resulting in a decrease in endogenous glutathione and neuronal death [56]. β-sheet-rich α-syn oligomers interact with lipid membrane, bringing about abnormal calcium influx and lipid peroxidation in an iron-dependent manner [57]. The process of cell death caused by α-syn-induced lipid peroxidation is called ferroptosis. The uptake of α-syn oligomers by SH-SY5Y cells destruct the cell membrane and ion homeostasis, giving rise to the activation of nitric oxide synthase (NOS), S-nitrosylation of key proteins, and change of cytoskeletal network, protein folding mechanism and ubiquitin proteasome system. All these effects induce cell apoptosis [58]. Externally added α-syn fibers can bind to the cytoplasmic membrane and act as nucleation sites, promoting oligomerization and internalization of α-syn and activating intracellular and extracellular apoptotic pathways [59]. These studies indicate the effect of α-syn oligomer on promoting apoptosis.

### 3.6. Inflammation

The α-syn pathology not only influences neurons, but also has effect on glial cells such as microglia and astrocytes. Astrocytes can uptake α-syn secreted by neuron and show an inflammatory response [60]. Oligomeric α-syn mainly interacts with toll-like receptors (TLR) to trigger inflammatory responses. Physiological concentration of α-syn oligomers can activate glial cells through a TLR4-dependent pathway, leading to the release of pro-inflammatory cytokines, including TNF-α, which in-turn cause neuronal death [61]. Besides, microglia could transform into a pro-inflammatory phenotype by oligomeric α-syn through TLR1 and TLR2 signaling pathway [62]. Another study also demonstrated that α-syn oligomer with particular conformations secreted by neurons is an agonist for TLR2 and causes an inflammatory response in microglia cells [63]. Notably, the molecular weight of α-syn oligomers used in these two studies are different, indicating the great diversity of oligomers. 

Compared to that from younger mice, microglia cells from adult mice exhibit a phagocytic deficiency for α-syn oligomers and increased TNF-α release, demonstrating that α-syn oligomers induced inflammatory response in PD pathology [64]. In a type 2 diabetes (T2D) model, MPTP treatment upregulates the level of oligomeric α-syn in both pancreas and midbrain, resulting in increased IL-1β secretion via NLRP3 activation, and ultimately exacerbates the loss of DA neurons [65]. The above studies linked inflammatory response to the pathology of PD, hinting that non-neuronal cells may also play an important part in PD pathogenesis and further research is needed to clarify this connection.

According to these studies, α-syn oligomers could impair DA neurons through a variety of mechanisms. This emphasizes that PD is not a disease with a single cause and suggests that different conformations of oligomers can bring about different toxic mechanisms. It is very important to classify the oligomers and clarify the properties of each group.

## 4. Prion Principle of α-syn Oligomers

Prions consist of misfolded prion protein (PrP) that can induce PrP monomers aggregate to PrP aggregates. In this way, prions can replicate themselves and spread to the entire brain, causing prion-protein diseases (PrD) such as Bovine spongiform encephalopathy (BSE) and Creutzfeldt–Jakob disease (CJD) [66]. PrD are currently the only confirmed neurodegenerative diseases transmitted by infection. However, all the misfolded proteins in neurodegenerative diseases show prion-like seed effects, including α-syn [67]. Direct evidence is that α-syn–positive Lewy bodies are detected in the grafts of PD patients who received embryonic midbrain cell transplantation, indicating the existence of host-to-graft transfer of α-syn pathology [68].

The oligomeric Ser129-phosphorylated (pS129) α-syn, but not insoluble aggregates, from brain tissues of dementia with Lewy bodies (DLB) cases exert a seed effect by converting recombinant monomers into fibers [69]. It is reported that an on-pathway α-syn oligomer with antiparallel β-sheet structure shorten the time for the monomers to form fibers [23]. However, a study suggested that two of three types of α-syn oligomers could enter cultured cells, promoting the aggregation of cytosolic α-syn [70]. The result illustrates that the seed effect of α-syn oligomers may rely on certain conformations [71]. Another study found that extracellular α-syn oligomers can cross cell membrane and promote intracellular α-syn oligomerization. Importantly, it mentioned that oligomeric α-syn can be translated into different oligomers according to brain environmental conditions, explaining the diversity of oligomers [72]. These studies demonstrate that oligomers can enhance the aggregation of α-syn.

As for the spread of α-syn from cell to cell, an experiment using α-syn-expressing SH-SY5Y cell model showed that α-syn can be secreted by exosomes in a calcium-dependent pattern. Notably, adding oligomer-interfering compounds to the media protected the recipient cells from neurotoxic effects, showing the role of oligomers in the spread of α-syn [73]. Hsp 70, a protein that can interact with extracellular α-syn, reduces the formation of α-syn oligomers, which could be ingested by nearby neurons and cause related toxicity [74]. A recent study suggested that oligomerization of α-syn occurs at the pre-synapse and oligomeric α-syn has the ability to spread from cells to cells in vivo [75]. In addition to neurons, microglia were also involved in the intercellular transmission of α-syn oligomers, by secreting oligomers through exosomes and inducing α-syn accumulation in neurons [76]. While the exact mechanism of α-syn oligomers intercellular transfer is unclear, the gap junction protein connexin-32 (Cx32) was proved to preferentially ingest α-syn oligomers in neurons and oligodendrocytes [77].

Besides, α-syn spreads not only in regions assuredly involved in PD, but also from the olfactory bulb (OB) or the gut to the brain. α-syn oligomers, rather than fibrils, can be uptake by OB neurons and spread to other regions of the brain [78]. There is evidence that all forms of α-syn including oligomers can be transported via the vagus nerve to the dorsal motor nucleus of the vagus nerve in the brain stem [79], supporting the hypothesis that PD pathology starts in enteric nervous system and diffuses to the brain [80].

Despite producing self-aggregation, α-syn can trigger the aggregation of other proteins that are involved in neurodegenerative diseases, such as amyloid-β (Aβ) and tau. This process is called “cross seeding”. In vivo, adding exogenous α-syn oligomers accelerated oligomerization of tau, leading to cell death [81]. According to an earlier study, the formation of β-sheet-rich neurotoxic tau oligomers was observed in vitro due to the cross-seeding effect of α-syn oligomers [82]. In addition, α-syn can form oligomers together with other amyloid proteins, which is called co-oligomerization [83]. Co-oligomers are generally more likely to be formed than self-oligomers at equilibrium. Interestingly, oligomeric Aβ and tau could also prompt the aggregation of α-syn, hinting the impact of interactions of amyloid proteins in neurodegenerative diseases [84,85]. These findings demonstrate the ability of α-syn oligomers to reproduce and propagate themselves, indicating the importance of blocking α-syn transmission in the treatment of PD.

## 5. Therapy Targeting α-syn Oligomers

α-syn oligomers can produce neurotoxicity in a variety of ways and have the ability to self-replicate and propagate from cell to cell. Since oligomeric α-syn takes an essential part in driving PD pathogenesis, it can be a potential therapeutic target. Reducing the production, promoting the clearance or blocking the spread of α-syn between cells may be helpful to alleviate the phenotype of PD (Figure 4) [86].

### 5.1. Natural Small Molecule Compounds

Some small molecule compounds can suppress the neurotoxicity of α-syn oligomers via different mechanisms. Squalamine, a natural compound known to have anticancer and antiviral activity, replaces α-syn from the surface of lipid vesicles and blocks the α-syn aggregation process [87]. Meanwhile, squalamine inhibits the interactions between α-syn oligomers and lipid membranes so that their toxicity is dramatically ameliorated [87]. Similarly, a naturally-produced aminosterol called trodusquemine protects human neuroblastoma cells from toxicity through displacing α-syn oligomers from cell membranes [88]. Baicalein, a flavonoid compound, could prevent the formation of α-syn oligomers in vitro and rescue motor deficits in a PD mouse model [89]. Nordihydroguaiaretic acid (NDGA) analogs enable α-syn to form α-helix structures that are necessary to perform physiological functions and prevent it from forming aggregates with toxic conformations [90]. Epigallocatechin gallate (EGCG), a component from green tea, displays its neuroprotective effects by promoting the production of non-toxic off-pathway oligomers and facilitating toxic oligomers to form less-toxic fibers [91]. These results imply that some natural small molecule compounds can reduce the level of toxic α-syn oligomers.

### 5.2. Heat Shock Proteins

Heat shock proteins (Hsps) are a family of proteins for which the expression is increased when cells are under stress. Many of them act as chaperones to help regulate the folding of proteins [92]. The co-expression of CHIP (carboxyl terminus of Hsp70-interacting protein), a multidomain chaperone, selectively degrades oligomeric α-syn and mitigates related toxicity [93]. Hsp70 alone performs a similar function [74]. Small molecules that act as Hsp90 inhibitors are also sufficient to prevent the formation of α-syn oligomers [94], indicating that Hsps take different part in the process of α-syn oligomerization. Hsp110 is capable of reducing α-syn aggregation both in vitro and in vivo. Furthermore, Hsp110 could diminish the propagation effect caused by exogenous α-syn aggregation seeds, exerting neuroprotective effects [95]. Hsp2 is also proved to be able to limit oligomerization of α-syn and degrade misfolded aggregates [96]. These studies demonstrated the role of Hsps in preventing α-syn from misfolding and their therapeutic potential in treating PD.

### 5.3. Oligomer-Specific Antibodies

Adaptive immunity plays an important role in PD pathology. Antibodies targeting α-syn facilitate the clearance of α-syn aggregates and may be an effective therapeutic strategy [97]. However, in many studies, antibodies generated through active immunization or administered via passive immunization are able to bind all types of α-syn, including monomers, oligomers and fibrils. To make this potential treatment more precise, it is necessary to develop antibodies targeting to certain type of α-syn [98,99]. Several antibodies targeting α-syn oligomers have been reported. These antibodies do not recognize the linear epitopes, but the conformational epitopes of α-syn, so they only recognize the aggregated α-syn with particular conformations [100]. mAb47, an α-syn oligomer-selective antibody, enters human neuroglioma cells through Fcγ receptors and neutralizes toxic α-syn aggregates in cells [101]. PD transgenic mice treated with an α-syn antibody that preferentially binds oligomeric species show reduced intracellular α-syn aggregation and lower extracellular oligomer level [102]. Autophagy induced by passive immunity could degrade α-syn in cells. Three kinds of antibodies, Syn-O1, -O2 and -F1, which recognize different species of α-syn, are able to decrease the accumulation of α-syn oligomers and prevent neurodegeneration in vivo [99]. A single chain antibody fragment (scFv) syn-10H specifically recognizes α-syn oligomers presented in PD [98]. This scFv can be useful as a potential diagnostic and therapeutic tool. Another scFv W20 recognizes the universal conformational epitope of amyloid oligomers and rescues motor and cognitive defects in PD and Huntington’s disease (HD) mouse models [103]. These studies suggest that oligomer-specific antibodies neutralize and depolymerize α-syn oligomers and can be developed as a therapeutic strategy.

### 5.4. Molecular Tweezer and Aptamers

A molecular tweezer named CLR01 is capable of inhibiting abnormal aggregation of amyloid protein and related toxicity. By binding with Lys residues, CLR01 disturbs hydrophobic and electrostatic interactions and nucleation is affected [104]. CLR01 is proved to reduce α-syn load in vivo and improve motor deficits by lessening the level of oligomeric α-syn [105]. Aptamers are short nucleic acids that recognize specific protein targets [106]. A 58-base DNA aptamers which was screened by systematic evolution bind α-syn with a high affinity, inhibiting the α-syn oligomerization in vitro and promote degradation of α-syn by lysosomal pathway [107]. These studies indicate that molecular tweezers and aptamers perform neuroprotective effects by interplaying with α-syn to prevent oligomer formation.

### 5.5. Receptor Antagonist

Some studies have identified the receptors to which α-syn oligomers bind, thus blocking these receptors can help reduce the neurotoxicity caused by oligomers. It is clear that oligomeric α-syn activate heterodimer TLR1/2 and TLR2, triggering inflammatory response, cytokine release and subsequent neurotoxicity [62,63]. Using small molecule or antibody targeting to TLR1/2 or TLR2 alleviated inflammation and neurodegeneration [62,63,108]. Blockade of mGluR5-evoked phosphorylation of NMDAR reversed memory deficits in mouse model [54]. These studies indicate the importance of revealing receptors that α-syn oligomers bind to, so that treatment methods can be developed.

## 6. Conclusions and Future Directions

More than 200 years have passed since James Parkinson originally described a disease with the characteristic of resting tremor, paralysis and diminished muscle strength, known as PD [109]. However, the exact pathogenesis of this desperate disease remains unclear. This leaves no effective treatment for PD currently. Lewy bodies, which are easily found in pathological tissues, have been associated with PD [8]. Whether Lewy bodies are the culprit leading to PD, or just innocent bystanders, is still inconclusive [110]. Numerous studies highlight the role of α-syn oligomers on neurotoxic effects and undoubtedly bring new hope to the understanding of the pathogenesis of PD. There is abundant evidence that α-syn oligomers with particular conformations can impair neurons and glial cells by damaging organelles and synapses, disrupting protein homeostasis and inducing inflammation. Some therapies targeting oligomeric α-syn have shown obvious neuroprotective effects. Nevertheless, there are still many unsolved problems in the research of oligomers.

There is a wide range of oligomers, encompassing variety of molecular weights and structures. Different types of oligomers have different properties and may show distinct effects on PD pathology. However, most studies on oligomers have not characterized the species of oligomers in detail. It will be inspiring if further studies could identify a specific oligomer and clear its neurotoxic mechanism. Besides, among the great number of oligomers, which are the key types that advance the process of PD? Highly neurotoxic oligomers should be focused on. Another question worth considering is whether oligomers obtained by in vitro incubation are present in vivo, since in vitro incubation cannot simulate the complex cellular environment. Understanding the structure of oligomers is helpful to understand the properties of them and to develop therapeutic methods. However, due to the heterogeneity and instability of α-syn oligomers, the atomic structure of oligomers has not been reported so far. Further studies are required to gain insights into the high-resolution structure of α-syn oligomers.

While numerous studies show the importance of α-syn oligomers in the pathology of PD, this does not mean that other forms of α-syn, such as fibril, are innocent. α-syn fibrils lead to severe cell death and motor deficits after injection in rat brain [111]. This may be partly due to the seed effect of α-syn fibrils. Other evidence is that the intramuscular injection of fibrillar α-syn in mice developed a fast CNS α-syn inclusion pathology and can be applied to generated rapid-onset PD model [112]. Understanding the role of different forms of α-syn, including oligomers and fibrils, in PD pathology is conducive to a deeper understanding of PD.

## Figures and Tables

**Figure 1 ijms-21-08645-f001:**
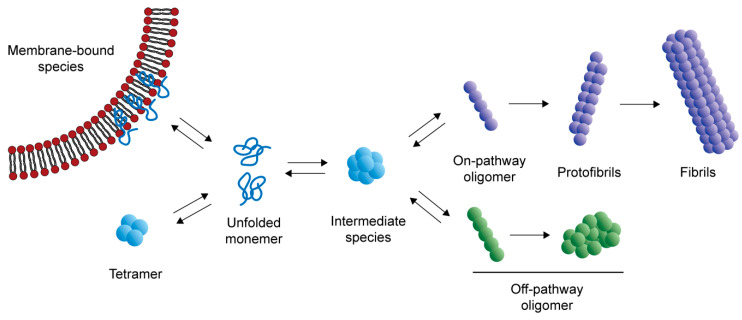
The process of α-syn aggregation. Under physiological conditions, α-syn exists as unfolded monomers, in equilibrium with membrane-bound species that promote SNARE-complex assembly and tetramers that can resist abnormal aggregation. When the balance between α-syn generation and clearance is disrupted, the monomers aggregate to form oligomers, including on-pathway oligomers and off-pathway oligomers. On-pathway oligomers tend to form protofibrils, and eventually fibrils. Other oligomers that cannot form amyloid fibrils are called off-pathway oligomers.

**Figure 2 ijms-21-08645-f002:**
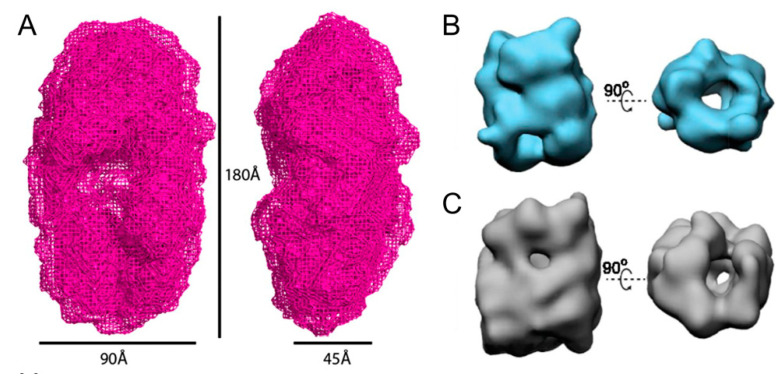
Current understanding of the structure of α-syn oligomers. (**A**) Structure of α-syn oligomers obtained by small angle X-ray scattering. The oligomer exhibits a 180 Å long and 90 Å wide circular shape with a cavity in the middle. (**B**,**C**) Structures of two subgroups of oligomers revealed by cryo-electron microscopy. Both subgroups have hollow cylindrical structures. (**A**) Adapted with permission from Reference [23], Proc. Natl. Acad. Sci. U.S.A. 2011, 108, 3246-3251. (**B**,**C**) Adapted with permission from Reference [24], Proc. Natl. Acad. Sci. U.S.A. 2015, 112, E1994-2003.

**Figure 3 ijms-21-08645-f003:**
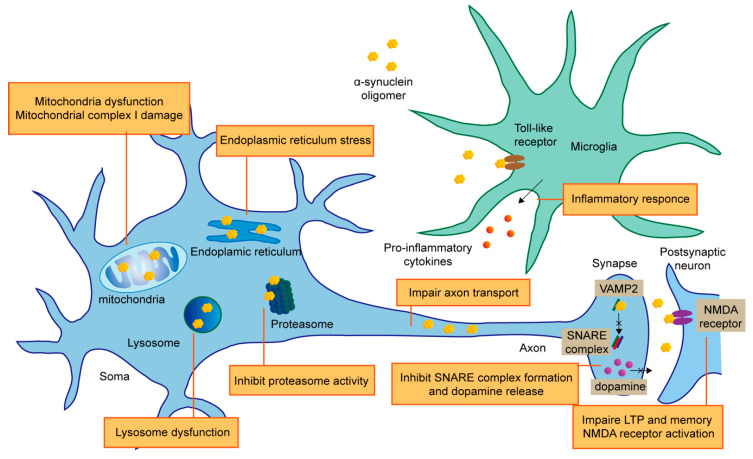
The mechanism of α-syn oligomers neurotoxicity. α-syn oligomers showing neurotoxic effects through multiple ways. Mitochondrial dysfunction can be triggered by α-syn oligomers through mitochondrial complex impairment and other kinds of mechanisms. Oligomeric α-syn uniquely lead to ER stress. The two main ways to maintain proteostasis, UPS and ALP, which are both affected by α-syn oligomers. Oligomers inhibit proteasome activity and cause lysosome dysfunction. α-syn is abundant in synapse while pathological α-syn oligomers damage synapse function via inhibiting SNARE complex formation and dopamine release. Some mutant α-syn oligomers have the ability to impair axon transport. α-syn oligomers also bind to NMDA receptor, resulting in membrane damage and LTP impairment. In addition to the direct cytotoxic effects on neurons, α-syn also participates in PD pathology by activating glial cells through toll-like receptor, giving rise to secretion of pro-inflammatory cytokines and neuron loss.

**Figure 4 ijms-21-08645-f004:**
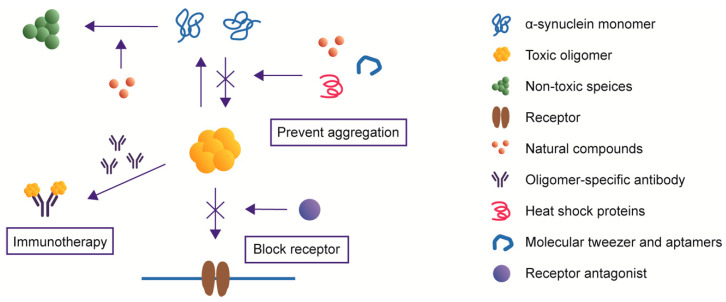
Therapies targeting α-syn oligomers. Natural compounds, heat shock proteins and molecular tweezers reduce the level of toxic oligomers. Oligomer-specific antibodies neutralize certain kinds of α-syn oligomers. Receptor antagonist prevents the oligomers from binding to the receptors and exhibiting cytotoxicity.

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
