# Peer review of "The Role of α-Synuclein Oligomers in Parkinson’s Disease"

_ijms, 2020, doi:10.3390/ijms21228645_

Round 1
Reviewer 1 Report
PD pathology is generally characterized by the progressive loss of dopaminergic neurons in SNpc and the presence of proteinaceous inclusion known as lewy body. It has been known that a-synuclein is involved in these cardinal neuropathological progressions in PD. Therefore, understanding underlying molecular mechanism of a-synuclein-mediated PD pathology is important for developing efficient therapeutic strategies for alleviating the PD phenotypes. Here, the authors well discuss about the cytotoxic functions of a-synuclein oligomers and prion-like aggregation and propagation of a-synuclein implying pathogenic mechanism of PD onset and progression. In addition, current applications of a-synuclein as therapeutic target of PD are also listed. I think that the overall manuscript is well written and informative in understanding the recent research trends related to synucleinopathy in PD field.
Author Response
Dear Reviewer,
We would like to thank you for your very helpful comments and your courage. Thank you very much for your time and effort.
Reviewer 2 Report
The topic of this review is very important and also the specific subtopics that the authors decided to exhamine in depth.
However, the organization of the article is not well done, there is a limited number of figures and the language must be improved.
I have underlined some mistakes and suggestions along the paper in the attached text

Author Response
Dear Reviewer,
We would like to thank you for your very helpful comments and suggestions. We have incorporated the suggestions one point by one in this revised manuscript and believe that your suggestions have significantly improved our manuscript, and hope that the revision we have made will address your concerns. Specific comments are addressed in details as follows. Please let us know if there is anything we can do to further address these questions or further improve the manuscript. Thank you very much for your time and effort.
Q: The organization of the article is not well done.
A: We modified some parts of the manuscript that are not well organized. We use the "Track Changes" function in Microsoft Word, so that changes are easily visible in the revised manuscript.
Some parts of the manuscript were reorganized to make it easier to understand (pg. 2, ln. 60, ln. 62-66; pg. 5, ln. 170-172).
In the comments in Figure 1, sentences that are not connected to the figure were removed (pg. 3, ln. 85-88).
The abbreviation “pS129” has been explained (pg. 7, ln. 254).
Q: There is a limited number of figures.
A: We added two figures to the manuscript. In the revised manuscript, Figure 2 describes current understanding of the structure of α-syn oligomers, as you suggested. And Figure 4 is a schematic diagram of therapies targeting α-syn oligomers.
Q: The language must be improved.
A: We read through the manuscript carefully, many typos and grammatical errors were corrected (pg. 1, ln. 11, 25-26, 42; pg. 2, ln. 49, 56, 59; pg. 3, ln. 115; pg. 4, ln. 125-126, 136, 145, 149; pg. 5, ln 177, 180, 200, 207; pg. 6, ln. 221; pg. 7, ln. 258-259, 265, 289; pg. 8, ln. 305-306, 325, 332, 338, 341; pg. 9, ln. 350, 361, 398).
Round 2
Reviewer 2 Report
The authors did not correct the form and the language. Please correct extensively the text
Author Response
Dear Reviewer,
We would like to thank you for your very helpful comments and suggestions. We have incorporated the suggestion in this revised manuscript and believe that your suggestion have significantly improved our manuscript, and hope that the revision we have made will address your concerns. Specific comments are addressed in details as follows. Please let us know if there is anything we can do to further address these questions or further improve the manuscript. Thank you very much for your time and effort.
Q: The authors did not correct the form and the language.
A: We have read through the manuscript carefully again. The form and the language were improved.